# Spatial Co-Clustering of Tuberculosis and HIV in Ethiopia

**DOI:** 10.3390/diseases10040106

**Published:** 2022-11-17

**Authors:** Leta Lencha Gemechu, Legesse Kassa Debusho

**Affiliations:** Department of Statistics, College of Science, Engineering and Technology, University of South Africa, Pretoria 0002, South Africa

**Keywords:** HIV, Moran’s I, bivariate Moran’s I, modified t-test, spatial co-clustering, tuberculosis

## Abstract

Background: Tuberculosis (TB) and HIV are epidemiologically associated, and their co-dynamics suggest that the two diseases are directly related at the population level and within the host. However, there is no or little information on the joint spatial patterns of the two diseases in Ethiopia. The main objective of the current study was to assess the spatial co-clustering of TB and HIV cases simultaneously in Ethiopia at the district level. Methods: District-level aggregated data collected from the national Health Management Information System (HMIS) for the years 2015 to 2018 on the number of TB cases enrolled in directly observed therapy, short course (DOTS) who were tested for HIV and the number of HIV patients enrolled in HIV care who were screened for TB during their last visit to health care facilities were used in this study. The univariate and bivariate global and local Moran’s *I* indices were applied to assess the spatial clustering of TB and HIV separately and jointly. Results: The results of this study show that the two diseases were significantly (*p*-value <0.001) spatially autocorrelated at the district level with minimum and maximum global Moran’s *I* values of 0.407 and 0.432 for TB, 0.102 and 0.247 for HIV, and 0.152 and 0.251 for joint TB/HIV. The district-level TB/HIV spatial co-clustering patterns in Ethiopia in most cases overlapped with the hot spots of TB and HIV. The TB/HIV hot-spot clusters may appear due to the observed high TB and HIV prevalence rates in the hot-spot districts. Our results also show that there were low-low TB/HIV co-clusters or cold spots in most of the Afar and Somali regions, which consistently appeared for the period 2015–2018. This may be due to very low notifications of both diseases in the regions. Conclusions: This study expanded knowledge about TB and HIV co-clustering in Ethiopia at the district level. The findings provide information to health policymakers in the country to plan geographically targeted and integrated interventions to jointly control TB and HIV.

## 1. Background

Tuberculosis (TB) is a contagious disease caused by the bacillus *Mycobacterium tuberculosis* (*Mtb*). Despite the tremendous efforts and encouraging progress obtained towards the control of the TB epidemic globally, it remains the single infectious disease that takes more lives each year, ranking above human immunodeficiency virus/acquired immunodeficiency syndrome (HIV/AIDS) until the coronavirus (COVID-19) pandemic spread worldwide [1]. In 2020, an estimated 9.9 million people fell ill with TB worldwide. This number shows a slight decline, about 0.87%, compared with 2019, and there were faster declines in the African (18%) and European regions (26%) [1]. HIV also continues to be a major global public health issue. According to UNAIDS estimates, globally, an estimated 37.7 million people were living with HIV in 2020 [2]; of these people, two-thirds of them lived in Sub-Saharan Africa (SSA). In 2020 and 2021, there were 1.5 million newly HIV-infected cases each year globally, and about 680,000 and 650,000 people died from AIDS-related illnesses in 2020 and 2021, respectively, worldwide [3,4]. In 2019 and 2020, there were an estimated respective 815,000 and 787,000 TB cases globally who were people living with HIV [3,5].

In 2019 and 2020, the estimated incidence rates for TB were 140 and 132 per 100,000 population, and the mortality rates were 21 and 17 per 100,000 population [1,6], respectively. An estimated 670,000 people in Ethiopia acquired HIV, and an estimated 12,000 people died of AIDS-related illness in 2019 [7]. The Government of Ethiopia has been taking several measures to control TB and HIV; for example, it established the National Tuberculosis and Leprosy Control Program (NTLCP) to harmonize the coordination and management of TB and leprosy at the country level in 1994 [8] and improved access to TB diagnostic facilities through its community-based Health Extension Program. Further, based on a WHO recommendation, the national responses for the integrated TB and HIV collaborative activities were updated in 2012 and aimed to reduce the burden of HIV among TB patients and the burden of TB among people living with HIV (PLHIV). Since then, the integrated TB and HIV service has expanded to the tertiary, secondary, and primary health care levels in Ethiopia [9]. Despite the measures that have been taken, Ethiopia remains among the 30 countries reported with a high burden of TB and HIV, TB/HIV co-infection, and multi-drug resistant (MDR)-TB from 2015 to 2020. The spatial analysis of a disease helps to understand the spatial epidemiology of the disease occurrence (incidence) at different administrative levels [10]. There are several studies in Ethiopia that have investigated the spatial clustering of TB and HIV separately at various levels [11,12,13,14,15,16,17,18,19]. Studies in other countries also reported spatial clustering of tuberculosis, e.g., in Brazil [20], China [21], Nigeria [22,23] and in South Africa [24]. Similar studies are also available for HIV, e.g., in Burundi [25], China [26], Malawi [27], Mozambique [28], South Africa [29] and Uganda [30]. Studies also show that an understanding of the spatial distribution pattern of TB (e.g., [31,32]) and HIV (e.g., [26,33,34,35,36]) helps to design focused interventions, such as facilitating access to TB control, community-based awareness creation and intervention, contact tracing and equitable access to treatment for vulnerable populations. HIV and TB are epidemiologically associated [37]. Observed co-dynamics suggest that the two diseases are directly related at the population level [38] and also within the host [39]. A study in Ethiopia also suggests that TB incidence is high in areas where HIV is highly prevalent [40]. Studies on the prevalence of TB-HIV co-infection have shown that the co-infection varies widely in Ethiopia (see, e.g., [40,41]) and has geographical clustering [42]. Similar observations have been made in other countries (see, e.g., [43,44,45]). There are studies in Ethiopia that have reported the epidemiology of TB-HIV co-infection at the hospital level [46,47]. Except for [42], the other studies include either a single district or region or only public health facilities in a district or a town. However, Alene et al. [42] have reported the spatial distribution of TB-HIV co-infection at the national level using three years of aggregated data. Despite the relevance of TB-HIV co-infection, there are still few publications on its spatial co-clustering, at least using yearly data at a national level. To design the most effective strategies that help to reduce the TB and HIV transmission rates, it is essential to have a more in-depth analysis of the epidemiological patterns of TB-HIV co-infection at the district level because targeting high-risk areas with effective control measures yields good results in controlling the pandemic [48]. The results of such an analysis could be used to inform locally targeted disease-specific or integrated control measures. In addition, knowledge of hot-spot and cold-spot (or high- and low-burden) areas is required for successful surveillance programs and optimal resource allocation [49]. Therefore, the main objective of the current study was to assess the spatial co-clustering of TB and HIV in Ethiopia at the district level for a four-year period, 2015 to 2018. Further, we were interested in assessing the spatial clustering of TB and HIV separately for the same period. These objectives were addressed using the aggregated data, collected from the national Health Management Information System (HMIS), on the number of TB cases enrolled in DOTS and who were tested for HIV, and the number of HIV patients enrolled in HIV care who were screened for TB during their last visit to health care facilities.

## 2. Material and Methodology

### 2.1. Study Area

Ethiopia is located in the northeastern part of Africa. It shares borders with Sudan in the west, Eritrea to the north, Djibouti in the northeast, Somalia in the east and southeast, and Kenya in the south. The country occupies an area of approximately 1,127,127 square kilometers. Administratively, before 2020, the Federal Democratic Republic of Ethiopia is divided into nine regional states (Tigray; Afar; Amhara; Oromia; Somali; Benishangul-Gumuz; Southern Nations, Nationalities, and Peoples’ (SNNP); Gambella; and Harari) and two city administrations (Addis Ababa and Dire Dawa). Each regional state is further divided into zones, a zone into districts (called “woreda”), and a district into kebeles (sub-districts). With the devolution of power to regional governments, public service delivery is under the jurisdiction of the regional states. The regional health bureaus are responsible for the administration of public health, while the districts are responsible for the planning and implementation of services. Districts comprise a well-defined population within a delineated administrative and geographical area. Districts contain networks of primary health care units such as health centers, health posts, and a district hospital. Health-related data are compiled at the district level and are reported to the Zone Health Department, Region Health Bureau, and then to the Federal Ministry of Health [42].

### 2.2. Data Sources

The data that we used for this study are secondary data including all types of TB and HIV cases recorded separately at health facilities, compiled and aggregated at the district level, and reported to the national level. All TB cases registered enroll in directly observed therapy, short course (DOTS) and test for HIV, and all HIV patients in HIV care are also screened for TB during their follow-up period. We called these data in this paper TB and HIV data or cases, respectively. Both TB and HIV are notifiable diseases in Ethiopia. These diseases are collected from the health center by the district Health Office, compiled, and reported to the Federal Ministry of Health through the Health Management Information System (HMIS) every quarter [50,51]. These data were used to examine the spatial clustering of TB and HIV cases in Ethiopia for four years, from 2015 to 2018, giving particular emphasis on the joint spatial co-clustering of both diseases at the district level. Related to these data, district mapping shape files were obtained from the Central Statistics Agency (CSA) of Ethiopia.

## 3. Statistical Analyses

In this paper, spatial analyses were conducted to identify geographical clustering of the HIV and TB cases separately and their co-clustering simultaneously in Ethiopia at the district levels for each year from 2015 to 2018. For the spatial analyses, the district geographical boundaries were geo-referenced. They were linked to the district HIV and TB data, and choropleth maps were developed for visualization using GeoDa software version 1.2 [52]. Then, the global pattern analysis to study the global spatial autocorrelation for the prevalence of HIV and TB separately was conducted using the univariate global Moran’s *I* [53]. The two diseases’ prevalences were investigated simultaneously using the bivariate global Moran’s *I* tests. The univariate local Moran’s *I* [54] statistics were applied to identify the local spatial clusters of HIV and TB separately [55], whereas the bivariate local Moran’s *I* [56] statistics were applied to investigate the simultaneous occurrence of both diseases and hence to identify the co-clustering of HIV and TB in Ethiopia for each year in the study period [44]. The significance of Moran’s *I* statistics is assessed by the Monte Carlo randomization technique. The spatial relationships of the districts were defined using a spatial weight matrix, and neighbourhoods were defined using Queen’s contiguity, where neighbours’ districts are defined as districts sharing borders or a common vertex with each other. The weight matrices for the local indicators of spatial association (autocorrelation) or for local Moran’s *I* statistics were defined using the nb2listw function from the spdep [57] R package.

In the choropleth map, for a single disease case, the local indicators of spatial analysis (LISA) classify districts into five clustering categories to show the spatial clusters and spatial outliers: high-high (hot-spot), which are districts with a high density of disease notification compared to the expected cases given a random distribution of disease; low-low, which are districts with a low density of disease notification (high-high and low-low suggest clustering of similar districts); low-high, which are districts with a low notification of disease sharing borders with districts with high notifications; high-low, which are districts with a high number of notified cases sharing borders with districts with low notifications; and not significant clustering. For the bivariate case, the software produces cluster maps in which the significant districts are classified into high-high, and low-low, and to describe the nature of spatial outliers of the two disease distributions at the district level, the districts are also classified as high-low and low-high. In each of the four classifications, the first attribute corresponds to the total number of TB cases, and the second corresponds to the total number of HIV cases in the neighbouring districts.

### Ethical Consideration

Permission for the study was obtained from the School of Science Ethics Committee, University of South Africa (ERC Reference Number: 2021/CSET/SOS/045). In addition, permission to use the data for this study was obtained from the Ethiopian Ministry of Health Office. In this study, we have used aggregated district-level data where individual patient information was not available; therefore, informed consent was not obtained from the study participants.

## 4. Results

### 4.1. Descriptive Statistics

The number of HIV-positive clients enrolled in HIV care who were screened for TB and the number of tuberculosis cases enrolled in Directly Observed Therapy Short Course (DOTS) in Ethiopia are given by the year throughout the study period for all regions and two city administrations in Appendix A. The numbers of HIV-positive clients enrolled in HIV care who were screened for TB per year in 2015, 2016, 2017, and 2018 were 623,926, 914,987, 104,328, and 719,651 (Appendix A), respectively. The average number of HIV-positive clients enrolled in HIV care who were screened for TB per year over the study period in Ethiopia was 824,973. The number of tuberculosis cases enrolled in DOTS and who were tested for HIV in 2015, 2016, 2017, and 2018 were 91,382, 91,692, 93,302, and 52,411, respectively (Appendix A); the average number of tuberculosis cases enrolled to DOTS and who were tested for HIV per year over the study period in Ethiopia was 82,197. The above results demonstrate that TB and HIV incidences using the study data are geographically heterogeneous over time, and this will be further explored in the next sections.

#### 4.1.1. Global Pattern Analyses

The results in Table 1 show that in each year within the study period, there was evidence of spatial clustering of HIV and TB in Ethiopia, i.e., the results indicated a significant spatial autocorrelation at the district level for both diseases. In each study year, the global Moran’s *I* test found significant positive autocorrelation (*p*-value <0.001) in HIV and TB cases. Observe from Table 1 that TB was more spatially correlated than HIV consistently over the study period. The positive values of global Moran’s *I* in Table 1 for each of the diseases show that any two spatial neighbouring districts tend to have similar prevalence rates for the study period.

The global bivariate Moran’s *I* was also consistently positive across the years, suggesting that the observed notification of HIV was positively associated with the notification of TB in the neighborhood districts. The annual Moran’s *I* values for TB, HIV, and the co-clustering of both diseases were relatively stable from 2015 to 2018; however, they reached a peak in 2018 for TB and in 2016 for HIV and the co-clustering of HIV and TB. The lowest values were observed for TB, HIV, and their co-clustering in 2015.

#### 4.1.2. Spatial Clustering of TB and HIV

The local indicators of spatial analysis (LISA) results show that the number of high-high (hot-spot districts) for 2015, 2016, 2017, and 2018 were 55, 40, 40, and 73, respectively. The highest number of districts that were identified as hot spots in 2015, 2016, 2017, and 2018 were in the Oromiya (n=25; 45.5% of the total hot spots), Oromiya (n=19; 47.5%), Oromiya (n=18; 45.0%) and Amhara (n=51; 69.9%) regions, respectively. A list of high-high and low-low clusters, as well as low-high and high-low spatial outliers, are presented in Appendix A, respectively. All 10 districts from the Addis Ababa city administration, 3 districts (Bahir Dar Zuriya, Gondar Zuriya, and Mirab Armacho) from the Amhara region, only the Legehare area from the Dire Dawa city administration, 4 districts (Adama, Burayu, Sebeta Hawas, and Shashemene) from the Oromiya region, and only the Wendo Genet district from the SNNP region were identified as statistically significant and consistent hot-spot districts for all study years (Figure 1, Appendix A). Furthermore, the Dega Damot and North Achefer districts from the Amhara region for 2015, 2017, and 2018; Dire Dawa town for 2015, 2016, and 2017; the Adola, Arsi Negele, and Mana districts from Oromiya for 2015, 2016, and 2017; and Shebedino from the SNNP region for 2015, 2016 and 2017 were identified as hot-spot districts (see Appendix A for the full list of hot-spots).

Except for the Addis Ababa city administration and the Tigray region, some districts were identified as low-low clusters in the Dire Dawa city administration and the other eight regions. The number of low-low districts which were statistically significant and consistently low-low clusters in all years varies from 1 (in the Dire Dawa city administration) to 35 (in the Somali region). For the full list of low-low clusters, see Appendix A. Similar to the low-low clusters, low-high and high-low spatial outliers were not found in all regions and city administrations; for example, in 2015, low-high clusters were found in the Amhara, Oromiya, and SNNP regions and the Dire Dawa city administration; in 2018, only a few high-low clusters were found in the Amhara, SNNP and Somali regions (Figure 1, Appendix A). Only Merawi town in the Amhara region and Bereh district in the Oromiya region had significant and consistent low-high clusters in all years, while only Este town in the Amhara region had high-low clusters (Figure 1); see Appendix A for a complete list of identified low-high and high-low spatial outliers for the years 2015 to 2018.

Similar to TB cases, the maps displayed in Figure 2 show that the LISA analysis identified hot spots and outliers for the notification of HIV cases in all districts of the Addis Ababa city administration and the Legehare area in the Dire Dawa city administration from 2015 to 2018, in twelve districts in the Amhara region (three in North Gondar, one in South Gondar, three in South Wolo, three in North Wolo and one in the Awi zone), one in Oromiya region (in Adama—the East Shewa zone) and one in Tigray region (Humera town—the Western Tigray zone) in 2015 (Figure 2, Appendix A). In 2016, hot spots were identified in ten districts of the Amhara region (three in North Gondar, six in South Wolo, and one in the Awi zone) and three districts of the Oromiya region (Adama, Sululta in Finfine zuria, and Burayu); in 2017, how spots were identified in seven districts in the Amhara region (three in the North Gondar and four in the South Wolo zones), and three hot-spots were identified in the Oromiya region (Adama, Sululta in Finfine Zuria and Burayu). In 2018, twenty-two identified districts in the Amhara region (three in North Gondar, one in South Gondar, eleven in South Wolo, five in North Wolo, one in Awi, and one in the Oromiya zone), the Adama district in the Oromiya region, and two districts in the Tigray region (Ambalage and Alamata, both from South Tigray) appeared as HIV hot-spots (Figure 2, Appendix A).

The number of low-high spatial outliers or discordant HIV clusters was almost equal in the study years; fourteen districts appeared as discordant HIV clusters (one in Afar, nine in Amhara, two in Oromiya and two in the Tigray region) in 2015; eight appeared in Amhara, one appeared in the Dire Dawa city administration and five appeared in the Oromiya region in 2016, and for 2018, there were one in Afar, seven in Amhara, four in Oromiya and two in Tigray region (Appendix A). For 2017, thirteen districts appeared as low-high HIV clusters or spatial outliers, with seven in Amhara, one in Harari, and five in the Oromiya region (Figure 2, Appendix A). Three towns in the Amhara region in 2015, six districts/towns in the Oromiya region in 2016, nine districts in the Oromiya region and only the Legehida district in Somali region appeared in 2017, and only two towns (Este in the Amhara region and Bedele in the Oromiya region) appeared as high-low spatial outliers for 2018 (Figure 2, Appendix A).

### 4.2. Spatial Co-Clustering of HIV and TB

To study the simultaneous occurrence or co-clustering of HIV and TB cases in Ethiopia at the district level for the study period, the bivariate local Moran’s *I* statistic was applied using the GeoDA software. The software produces a cluster map in which the significant districts are classified into high-high, low-low, high-low, and low-high. The results are presented in Figure 3.

In 2015, in total, there were 22 high-high TB and HIV occurrences and co-clusters (Figure 3, Appendix A), consisting of all ten districts of Addis Ababa city administration, ten districts/towns from the Amhara region and two districts from Oromiya. All ten districts of the Addis Ababa city administration, nine districts/towns of the Amhara region, Legehare area from the Dire Dawa city administration, and five districts from the Oromiya region appeared as high-high TB and HIV co-clusters for 2016. For 2017, the spatial patterns in TB and HIV high-high co-clusters were almost the same as for 2016. In 2018, there were 37 high-high TB and HIV occurrences and co-clusters, including all districts in the Addis Ababa city administration, nine of the districts for 2016, and fourteen more from the Amhara region, the Legehare area from the Dire Dawa city administration, and two of the Oromiya districts (Adama and Sebeta Hawas) and the Alamata district from the Tigray region appeared as hot-spots for TB and HIV (Figure 3, Appendix A).

Similar to TB LISA results, except in the Addis Ababa city administration and the Amhara and Tigray regions, there were low-low HIV and TB notification occurrences and co-clusters in districts of other regions. However, the number of the low-low co-clusters was unstable across the regions and varied over the study years (Appendix A). Their occurrences were high in Somali, SNNP, and Oromiya from 2015 to 2018, except in SNNP for 2017, where the number of low-low clusters was relatively low. The number of TB and HIV low-low co-clusters generally increased with time in the Oromiya region, decreased in the SNNP region until 2017, and reached its peak in 2018, but for the Somali region, the number was stable (Appendix A).

Fifty spatial outliers were identified for 2015; twelve of them were classified as low-high spatial outliers or as discordant TB and HIV co-clusters, which were distributed between the Amhara (ten districts) and Oromiya regions (two districts). Thirty-eight districts appeared as high-low spatial outliers, and they were from Amhara (two districts), Harari (two districts), Oromiya (twenty-four districts), SNNP (nine districts), and one district from the Somali region. For 2016, there were forty-one discordant clusters, of which seven (four from the Amhara region and three from the Oromiya region) appeared as low-high and thirty-four appeared as high-low (one each from the Amhara and Gambella regions, two each from the Harari and Somali regions, seventeen from Oromiya and eleven from the SNNP region). In total, thirty-four spatial outliers appeared in 2018 where thirteen were low-high (one from Afar, six from Amhara, and three each from the Oromiya and Tigray regions), while twenty-one districts (one from Afar, thirteen from Oromiya, five from SNNP, and two from the Somali region) appeared as high-low (Figure 3, Appendix A).

## 5. Discussion

Tuberculosis and HIV still represent the most important infectious diseases around the globe. Despite advances made in their eradication via WHO-coordinated efforts, these diseases still present alarming data on morbidity and mortality [1,2]. In the current paper, we have investigated the spatial clustering of TB and HIV separately and the spatial co-clustering of both diseases in Ethiopia at the district level for a four-year period from 2015 to 2018.

The findings of this study show that the occurrences of both diseases were geographically heterogeneous over time. These results are consistent with the findings of [42], whose results indicate that the prevalence of TB among people living with HIV and the prevalence of HIV among TB patients varied in Ethiopia at the district level, but the data that we have used are different from theirs. Several studies conducted in Ethiopia [11,12,14,15,16,18] and in other countries, including Nigeria [22,23], China [21], and South Africa [24], revealed a strong tendency toward the spatial clustering of TB at different geographic scales. Similarly, a number of studies showed the spatial clustering of HIV at different scales [19,25,26,27].

The results from LISA show that the notification of TB was strongly spatially clustered in the districts of the Addis Ababa city administration; the Gondar Zuriya, Bahir Dar Zuriya, and Armacho districts in the Amhara region; the Legehare area in Dire Dawa; the Adama, Burayu, Sebeta Hawas, and Shashemene districts in the Oromiya region and the Wendo Genet district in the SNNP region. However, the spatial pattern of TB showed changes in the eastern (in 2017 and 2018) and northern (in 2018) parts of the country. In 2018, the hot spots were more concentrated in the Amhara region; about 70% of the observed hot spots were in this region.

Generally, the TB hot spots in the country over the study period appeared in more urbanized areas such as Addis Ababa, Adama, Dire Dawa, Bahir Dar, and Shashemene. These findings are supported by the results of [13], which identified urbanization and population density as main risk factors for TB. Additionally, it may be related to the migration of people within districts or across neighboring districts for jobs and better living conditions. Studies in Ethiopia have indicated that the spatial clustering of TB is associated with migration [11,12,13,14,15,17], and ongoing TB transmission is also high in overcrowded and congested urban areas [10,13,14,17]. In addition, other studies in Ethiopia [15,18] and in other countries such as Argentina [58], Brazil [59] and China [60] show that TB incidence rates are associated with poor living conditions and housing. The LISA cluster maps illustrated that there were hot spot areas on the border of North Sudan over the study period (2015 to 2018) and Eritrea in 2018. These findings support the results of the existing literature [17,42]; therefore, they may affirm that there is a relationship between TB transmission and international border or territorial space. Hence, it is necessary to extend the country-level analysis to higher spatial dimensions that include at least neighbouring countries to obtain global solutions and targeted interventions [61,62,63].

As with the TB case, the results of the LISA analysis show the spatial clustering of HIV in Ethiopia at the district level over the study period (2015 to 2018). The high-high clustering or hot-spots of HIV significantly and consistently appeared in the districts of the Addis Ababa city administration, in three districts from the Amhara region (Ambasel, Dembia, and Gondar Zuriya), the Legehare area from the Dire Dawa city administration, and Adama from the Oromiya region. In addition to Addis Ababa, Dire Dawa, and Adama, the current study identified that the country’s main cities or towns, such as Woldiya, Humera, Kombolcha, Dese, and Kobo, appeared as HIV hot spots. This may be because such places have a higher number of commercial sex workers [64,65,66], and some of the cities/towns are transport corridors for truck drivers or long-distance vehicle drivers, e.g.,Adama, Woldiya, and Kobo for trucks coming or going from or to the Djibouti port [67]. In addition, cities, towns, or generally urban areas are associated with a high rate of HIV infection [68,69] as residents in urban areas have higher population movement due to labor, migration, and trading. A hot spot was observed at Humera town in 2015, which shares a border with North Sudan and Eritrea. This area is in the Tigray region agricultural center and the gateway to North Sudan and is used by returnees [64]. This HIV hot-spot may be related to agricultural activities; some studies suggest that border areas and agricultural activities are potential risk factors for infectious disease [16,68,70].

In the current study, the results revealed that TB was more spatially correlated than HIV in Ethiopia for each year in the study period. However, this finding contradicts the results of Aturinde et al. [44], who found that HIV was more spatially correlated than TB in Uganda for the period of 2015–2017. Possibly, this difference could be due to the high burden of HIV (6.4% prevalence rate) in Uganda compared to 0.9% in Ethiopia. The univariate global Moran’s *I* statistics for TB and HIV were positive, suggesting that neighbouring districts tend to possess similar characteristics in both disease prevalences. Similarly, the global bivariate Moran’s *I* statistic was positive for the study period, and this implies that the two diseases were positively influenced by neighbouring districts.

The simultaneous TB and HIV spatial co-clustering patterns in Ethiopia at the district level, in most cases, overlapped with the hot spots of TB and HIV. The TB and HIV spatial co-clustering for the study period therefore could be explained by reported findings on TB and HIV in the literature, as discussed in the above paragraphs. Our findings on TB-HIV co-clustering clearly indicated an almost similar trend for the study period; hot-spot areas occurred mostly in the central part of the country (including Addis Ababa) and some part of the northeast and northwest. Several studies conducted at the regional level in Ethiopia support our findings [40,41,46,47]. Additionally, the results of a systematic review and meta-analysis of TB-HIV distribution in Ethiopia from 2007 to 2017 [71] showed regional and small-scale spatial variation in TB-HIV, which agrees with our findings. Similarly, the 2016 National TB and Leprosy control report revealed regional variation in TB, HIV, and TB-HIV prevalence, identifying the Addis Ababa and Somali regions as having high and low prevalence rates of TB-HIV co-infection, respectively [72]. Our results also show that there were low-low TB and HIV co-clusters, or cold spots, in most of Afar and the Somali regions, which consistently appeared for the period 2015–2018. This may be due to very low notifications of both diseases in the regions.

Although there are few studies that have investigated the spatial clustering of HIV and TB separately at the national scale in Ethiopia [66] and TB/HIV co-infection, e.g., [42], to the best of our knowledge, this is the first spatial study on the co-clustering of TB and HIV using Ethiopian data. However, there were some limitations that could have affected our findings. First, the data were aggregated at the district level; therefore, the findings of this study cannot be representative of small administrative units of the country or kebele or household or individual levels. Second, since HIV and TB data were collected from the national HMIS electronic surveillance system, the reported HIV and TB cases might not reflect the actual burden of the diseases in a district due to the underreporting or underdetection of cases. For example, symptomatic individuals who did not receive HIV or/and TB diagnosis and treatment might remain unreported. Generally, the use of secondary data could affect the results of this study because of selection bias, information bias, or both. First, the data set includes those cases recorded only at health centers, which were compiled and reported to the districts, the zonal level, the regional level, and then the national level of the ministry of health offices. However, there may be unregistered TB and HIV cases mainly due to limitations of health service access, especially in rural remote areas. This missing data, called selection bias, could be one possible bias affecting the study results. Secondly, there may be missing data, alteration of data or systematic distortions when collecting information at any stage, and these could add another potential bias called information bias, which could affect the study’s findings. Third, the results could be more interesting if they were supported by spatial generalized linear models, specifically using count models to assess potential risk factors for TB, HIV, and both TB and HIV; however, these data were not available in the national HMIS.

According to recent studies, the outbreak of COVID-19 has become one of the contributing factors to the increase in morbidity and mortality related to TB cases. For instance, the 2020 WHO estimates using data from 84 countries indicate that the number of TB patients receiving care was reduced by 1.4 million in 2020 compared to 2019 (reduced by 21% from 2019). According to this WHO estimate, these COVID-19-related challenges in access to TB care could cause an additional half a million TB deaths. Findings from other studies [73,74,75] also showed strong association between TB and COVID-19 and between the mortality rates of the first cohort of patients with COVID-19 and TB co-infection, indicating the importance of considering their strong relationship in a future study.

Generally, our study results showed strong spatial co-clustering of TB-HIV at the district level. These spatial heterogeneities or clusterings may be due to different factors that should be studied further in detail. Therefore, based on our findings, we recommend that authorities should strengthen their intervention mechanisms, such as facilitating access to TB and HIV control, providing early screening and treatment, the introduction of community-based awareness creation, and allocating resources to hot-spot areas.

## 6. Conclusions

In this paper, we assessed the spatial clustering of TB and HIV separately and assessed their joint spatial clustering by applying the global and local Moran’s *I* statistics with LISA cluster maps. The results from applying these statistics showed that TB and HIV were strongly spatially co-clustered in Ethiopia at the district and regional levels for the period 2015–2018. The LISA results also showed the strong spatial co-clustering of TB and HIV in some specific areas during the study period. All districts of the Addis Ababa city administration, some districts in the Amhara region (north and south Gondar, south and north Wollo and west Gojjam zones), some main towns such as Adama and Shashemene in the Oromiya region, and Dire Dawa city were among some of the areas with TB and HIV hot-spots or co-clusters. Most districts of the Afar and Somali regions were significantly cold spots; however, the discordant districts were generally unstable over the study period, except for Este town in the south Gondar zone, which had relatively low notification rates of TB in the neighborhood of high prevalence rates of HIV. This study expanded knowledge about TB and HIV co-clustering in Ethiopia at the district level. It also allowed the identification of districts with high TB and HIV co-clustering cases, which should be a priority for control interventions. Furthermore, the findings of this study can provide valuable information to health policymakers in Ethiopia for the improvement of the national or regional responses for geographically targeted and integrated TB and HIV/AIDS collaborative activities and to strengthen actions for the prevention of TB-HIV co-infection.

## Figures and Tables

**Figure 1 diseases-10-00106-f001:**
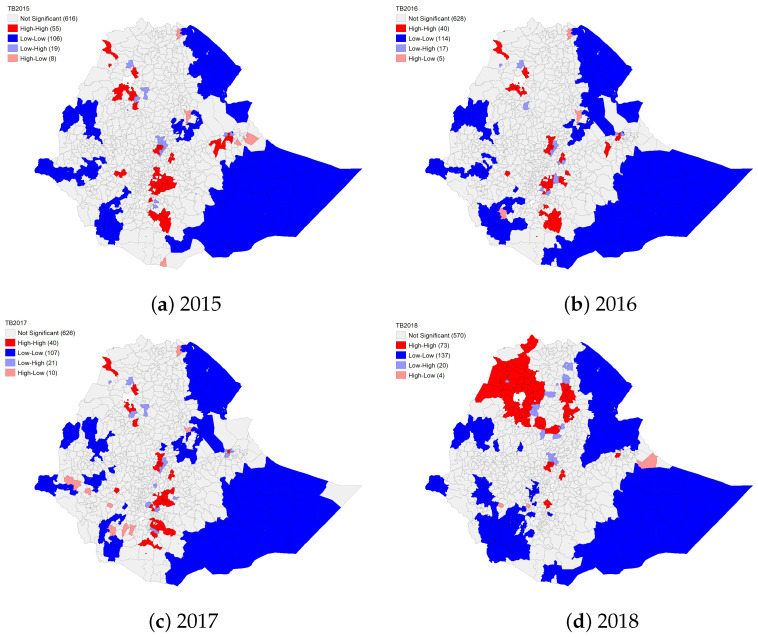
Spatial clusters and spatial outliers of tuberculosis notification in Ethiopia at district level from 2015 to 2018 using the local Moran’s *I* statistic.

**Figure 2 diseases-10-00106-f002:**
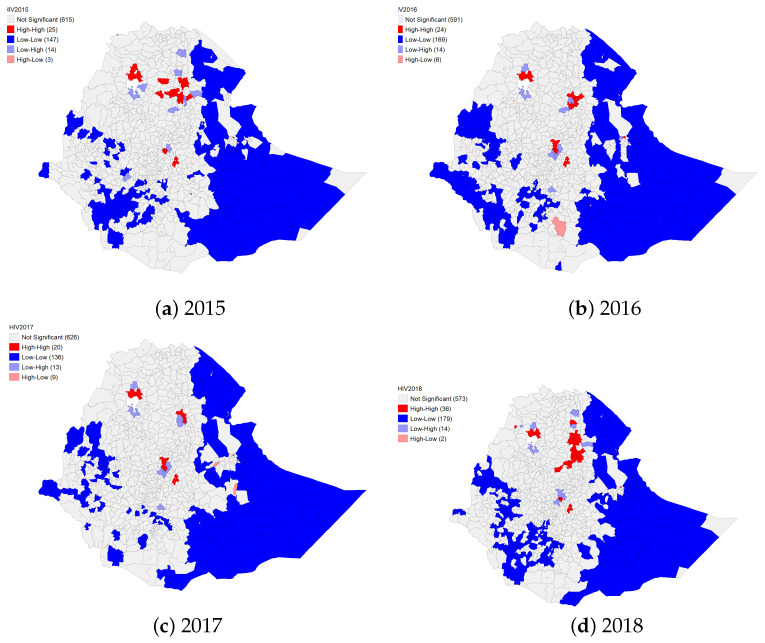
Spatial clusters and spatial outliers of HIV cases in Ethiopia at district level from 2015 to 2018 using the local Moran’s *I* statistic.

**Figure 3 diseases-10-00106-f003:**
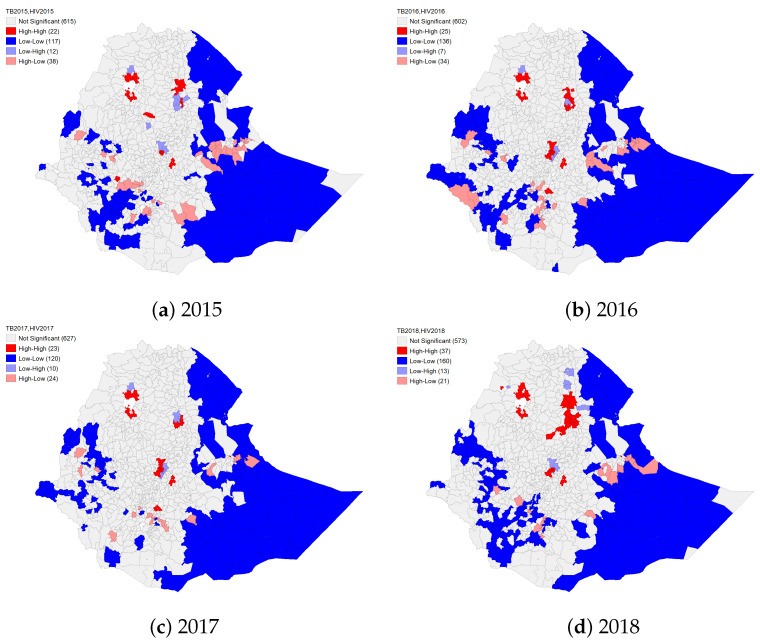
Spatial co-clustering of HIV and tuberculosis in Ethiopia at the district level from 2015 to 2018 using the binary local Moran’s *I* statistic.

**Table 1 diseases-10-00106-t001:** Moran’s *I* statistic and correlation (*p*-value) for HIV and TB cases by year.

	Year			
	2015	2016	2017	2018
Global Moran’s I				
Tuberculosis	0.407 (<0.001)	0.423 (<0.001)	0.411 (<0.001)	0.432 (<0.001)
HIV	0.102 (0.0036)	0.247 (<0.001)	0.238 (<0.001)	0.202 (0.0006)
Bivariate Global Moran’s *I*				
TB/HIV	0.152 (<0.001)	0.251 (<0.001)	0.230 (<0.001)	0.224 (<0.001)

## Data Availability

The data that support the findings of this study are available from the Ethiopian Ministry of Health Office, but restrictions apply to the availability of these data, which were used under a license for the current study and thus are not publicly available. Data are, however, available from the authors upon reasonable request and with the permission of the Ethiopian Ministry of Health Office.

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
