# Peer review of "Spatial Co-Clustering of Tuberculosis and HIV in Ethiopia"

_diseases, 2022, doi:10.3390/diseases10040106_

Round 1

Reviewer 1 Report

The manuscript entitled “Spatial co-clustering of tuberculosis and HIV in Ethiopia” report the spatial co-clustering of TB and HIV cases in Ethiopia, which is valuable for making focused control policies for these two diseases in this region.  However, several concerns have to be addressed before being accepted for publication.

Majors:

1.     The data collected in this study were the cases of TB screening for HIV and HIV cases testing for TB, by using which the authors conducted the analyses and drew conclusions on the spatial clusters of TB, HIV, and co-infection of the two diseases. To what extent did the data represent the real number of cases of the two diseases? Was it possible a biased conclusion and a higher estimated risk of TB-HIV co-clustering have been got because of the data source?

2.     What’s the prevalence of TB-HIV co-infection in different regions of Ethiopia? Are the spatial clusters obtained in this study consistent with that? This point should be discussed extensively in the contents.

3.     Although a summary of HIV and TB cases is presented in Table S1, the current study was to explore the spatial clusters of the two diseases at the district level, therefore, a table of the cases number of the two diseases summarized at which level is needed.

Minors:

1.     In this study, the authors didn’t estimate the prevalence of the two diseases, only the number of cases has been used for the analyses. Therefore, a thorough checking is suggested to assure that the two terms were used properly.

2.     Line 11, replace “districts” with “district”.

3.     Delete “were” in line 15.

4.     The title of Figure 2, uses “case” instead of “prevalence”.

5.     The title of Figure 3, should be “at district level”.

Author Response

Reviewer 1

Comments and Suggestions for Authors

The manuscript entitled “Spatial co-clustering of tuberculosis and HIV in Ethiopia” report the spatial co-clustering of TB and HIV cases in Ethiopia, which is valuable for making focused control policies for these two diseases in this region.  However, several concerns have to be addressed before being accepted for publication.

Majors:

  1. The data collected in this study were the cases of TB screening for HIV and HIV cases testing for TB, by using which the authors conducted the analyses and drew conclusions on the spatial clusters of TB, HIV, and co-infection of the two diseases. To what extent did the data represent the real number of cases of the two diseases? Was it possible a biased conclusion and a higher estimated risk of TB-HIV co-clustering have been got because of the data source?

Response:  

Thank you for the valuable comments and questions.

Health management information system (HMIS) organized under Federal Ministry of Health, Ethiopia (FMOH) is mainly responsible to stores, analyses, and evaluates health-related data collected from health facilities at district level. These data send by district authorities to zones, regions and then national HMIS.  Specific to TB and HIV, diagnosis and treatment are free at all public health centers in the country and the procedure of data collection is as follows: 

In both public and private health centers, a person infected with TB or HIV will be given the necessary treatment and registered for continuous follow-up. During this follow up periods TB patients will be tested for HIV and vice-versa. All TB and HIV-related data recorded at any health center will be compiled and reported to the district, zonal, regional, and national levels. The data that we have used in this study contains all TB and HIV annual aggregated case notifications at district levels for the period 2015-2018; collected and compiled using the above procedure. Therefore, we consider the data as a real number of cases notifications of the two diseases (TB and HIV) registered in Ethiopia during the study period with possible limitations related to any misinformation that may occur at any stage and unregistered cases due to limitation of access to health facility centers (the limitations of the study are discussed in Lines 368-381 on page 10).

  1. What’s the prevalence of TB-HIV co-infection in different regions of Ethiopia? Are the spatial clusters obtained in this study consistent with that? This point should be discussed extensively in the contents.

Response:

Though there is limited study conducted at both regional and national level, the existing literature on spatial co-clustering of TB and HIV in Ethiopia shows evidence of clustering at different administrative levels. Yes, the spatial clusters obtained in this study are consistent with some of the literature and this is discussed in Lines 355-364, page 9. Literature show that the prevalence of co-infection various from region to region (see the discussion in Lines 65-68, page 2).

  1. Although a summary of HIV and TB cases is presented in Table S1, the current study was to explore the spatial clusters of the two diseases at the district level, therefore, a table of the cases number of the two diseases summarized at which level is needed. 

Response:

Yes, the main aim of the study was to investigate the spatial co-clusters of TB and HIV at the district level and all results presented are based on annual aggregated data at district levels. Regional level summary data on the number of cases is provided to support comparison at regional level and the detail description of clusters with their respective districts are given in Tables S2-S7.  

Minors:

  1. In this study, the authors didn’t estimate the prevalence of the two diseases, only the number of cases has been used for the analyses. Therefore, a thorough checking is suggested to assure that the two terms were used properly.
  2. Line 11, replace “districts” with “district”.
  3. Delete “were” in line 15.
  4. The title of Figure 2, uses “case” instead of “prevalence”.
  5. The title of Figure 3, should be “at district level”.

Response:  

Thank you for your valuable comments. We have considered and corrected all the above points raised as minor comments in the revised version of the manuscript.  

Reviewer 2 Report

Congrats to the authors for completing this work. Here are my comments to further improve the draft - 

* Abstract: Conclusion - needs to be revised

* Background - Line 31 - Faster decline in African and European regions by how much?

* Line 53 - What is considered in the spatial analysis of a disease? Incidence? Prevalence? Please mention

* Line 61 - Focused interventions such as?

* Line 76 - in "cold spots", how do you differentiate between true low burden and a lack of reporting?

* Materials - Were all states and districts included in the study? Please mention

* Lines 98-102: Administrative data before 2020 needs to be replaced by current data

* Moran I or Moran's I? Use one form consistently throughout the draft

* Results: Text needs to be condensed. Suggest mentioning names of places in the Figure legend and not in the text

* Discussion - Line 311 - Address comments

* Line 351 Suggest reasons for the contradicting results

Best wishes,

Author Response

Reviewer 2

Comments and Suggestions for Authors

Congrats to the authors for completing this work. Here are my comments to further improve the draft - 

 Abstract: Conclusion - needs to be revised

Response:

Thank you, we have considered your comments and revised our conclusion.

* Background - Line 31 - Faster decline in African and European regions by how much?

Response:

The decline percentage was 26% in European WHO region and 18 % in African region. We have added this information in the revised version in Line 31 on page 1.

* Line 53 - What is considered in the spatial analysis of a disease? Incidence? Prevalence? Please mention

Response:

It is case notification / incidence that was considered.

* Line 61 - Focused interventions such as?

Response:

According to the study findings the recommended Focused intervention techniques were facilitating access to TB control, introduction of active case finding, community-based awareness creation and intervention, study-based resource allocation, contact tracing and equitable access to treatment for vulnerable populations (this was mentioned in Lines 61-64, page 2 in the revised version of the manuscript).

* Line 76 - in "cold spots", how do you differentiate between true low burden and a lack of reporting?

Response:

Since the data are reported from health facility to national HMIS via health authority channel, it is expected that reported figure is exact number of cases in each district.  However, under reporting might be occurred at some locations due to lack of nearby health center or mismanagement at any stages. Therefore, we considered cold spot areas as low diseases burden taking in to account the above limitation.

* Materials - Were all states and districts included in the study? Please mention

Response:

Yes, all districts and regions located in Ethiopia were considered in the study.

* Lines 98-102: Administrative data before 2020 needs to be replaced by current data

Response:

We used 2020 administrative data, because the data used in the study was aggregated in district name and locations (areas) defined during the study period till 2020. There were some districts merged and split after 2020.

* Moran I or Moran's I? Use one form consistently throughout the draft

Response:

Moran’s I, we have considered the comment and made correction in the revised version of the manuscript.

* Results: Text needs to be condensed. Suggest mentioning names of places in the Figure legend and not in the text.

Response:

We have considered the comments and reduced the text in result section, specifically we have removed name of hot and cold spot districts and replace by large geographically areas cluster like regional and number of districts at zonal level.  

* Discussion - Line 311 - Address comments

* Line 351 Suggest reasons for the contradicting results.

Response:

Possible strong spatial clustering of HIV occurred in Uganda is due to high burden of HV (6.4% prevalence rate) compared to 0.9% in Ethiopia. This was mentioned in Lines 346-347 on page 9 in the revised version.

Reviewer 3 Report

In the manuscript, the authors proposed an interesting and important study of spatial correlations between people infected with HIV and tuberculosis in Ethiopia. The statistical methods are adequate and well-referenced, and the results and discussions are relevant and suitable. Some minor revisions, mainly on the references section must be done, however, the work can be published after minor revisions as listed below.

 In the Abstract I suggest to the authors present just qualitative parameters. Quantitative ones are not adequate to present in this part of the text. Please, prefer qualitative descriptions.

 The introduction is well-written and instigates the spatial analysis of tuberculosis and the synergism with other sexual transmitting diseases, such as HIV. The authors focused the intro on a conclusive panorama, with an excellent review of the literature. Well-done.

 The methodology is understanding and well-referenced.

In the results section, I suggest to the authors describe in figures 1-3 at least the main clusters' names, to take easy the reading of it.

 The discussion is very well. Congratulations to the authors.

 References need to be better formatted and entirely reviewed. The references do not follow instructions to the authors and must be entirely revised. References to electronic websites must contain the date and must be well-conducted. Most references do not have a number of pages, issues, or even formatting according to instructions to authors. Please review it.

 The supplementary information presented by the authors is a good font of data, making the manuscript data faithful and well-constructed. However, the authors must increase the quality of the tables and data presented, formatting all the documents presented there. So, I strongly suggest reviewing the formatting of all supplementary data.

 The manuscript can be published in Diseases journal after minor revisions.

Author Response

Reviewer 3

Comments and Suggestions for Authors

In the manuscript, the authors proposed an interesting and important study of spatial correlations between people infected with HIV and tuberculosis in Ethiopia. The statistical methods are adequate and well-referenced, and the results and discussions are relevant and suitable. Some minor revisions, mainly on the references section must be done, however, the work can be published after minor revisions as listed below.

In the Abstract I suggest to the authors present just qualitative parameters. Quantitative ones are not adequate to present in this part of the text. Please, prefer qualitative descriptions.

The introduction is well-written and instigates the spatial analysis of tuberculosis and the synergism with other sexual transmitting diseases, such as HIV. The authors focused the intro on a conclusive panorama, with an excellent review of the literature. Well-done.

The methodology is understanding and well-referenced.

In the results section, I suggest to the authors describe in figures 1-3 at least the main clusters' names, to take easy the reading of it.

The discussion is very well. Congratulations to the authors.

References need to be better formatted and entirely reviewed. The references do not follow instructions to the authors and must be entirely revised. References to electronic websites must contain the date and must be well-conducted. Most references do not have a number of pages, issues, or even formatting according to instructions to authors. Please review it.

The supplementary information presented by the authors is a good font of data, making the manuscript data faithful and well-constructed. However, the authors must increase the quality of the tables and data presented, formatting all the documents presented there. So, I strongly suggest reviewing the formatting of all supplementary data.

The manuscript can be published in Diseases journal after minor revisions.

Response:

Thank you for the constructive comments.

  • We have considered all the comments and update the abstract, reduce result discussion considering main cluster areas like regional and exceptional cases.
  • We have corrected the reference format and supplementary information as per the comment in the revised version of the manuscript.

Reviewer 4 Report

The study entitled “Spatial co-clustering of tuberculosis and HIV in Ethiopia” expands the knowledge about TB and HIV co-clustering in Ethiopia at district level. This is a useful study. However, there are some points that will improve the quality of the manuscript.

1.      There are some grammar, formatting/spacing issues that must be resolved by the authors.

2.      There are some unnecessary references in the introduction part. The authors can remove those references.

3.      The objective of the study must be provided at the end of the introduction part.

4.      The author must explain why did they not include the data from 2019-2021 in this study.

5.      There is no need for the text of the line 88-92.

6.      The inclusion and exclusion criteria need more clarification.

7.      Please mention the software used for the statistical analysis.

8.      Th authors must compare the results of similar studies reported in Ethiopia and Africa region. Please refer the reference 10-18, 21-23, and 37 of the manuscript.

9.      The authors must discuss the effect of COVID-19 pandemic on the epidemiology of the TB.

10.  A separate paragraph discussing the recommendation based on the result must be mentioned at the end of the discussion.

Author Response

Reviewer 4

Comments and Suggestions for Authors

The study entitled “Spatial co-clustering of tuberculosis and HIV in Ethiopia” expands the knowledge about TB and HIV co-clustering in Ethiopia at district level. This is a useful study. However, there are some points that will improve the quality of the manuscript.

  1. There are some grammar, formatting/spacing issues that must be resolved by the authors.

Response:

Thank you for the comments

  • We have considered the comments and correct grammar, format and spacing issues in the revised version of the manuscript.
  1. There are some unnecessary references in the introduction part. The authors can remove those references.

Response:

  • The authors believe that all used references are relevant to the current study, therefore we did not remove any reference.
  1. The objective of the study must be provided at the end of the introduction part.

Response:

  • The objectives now appear at the end of the introduction section in Lines 82-88 on page 2.
  1. The author must explain why did they not include the data from 2019-2021 in this study.

Response:

  • In this study, we have used the most updated data provided by the Ethiopian Ministry of Health and can’t get the access of recent data from 2019-2021.
  1. There is no need for the text of the line 88-92.

Response:

  • We have considered the comments and removed the suggested text in the revised version of the manuscript.
  1. The inclusion and exclusion criteria need more clarification.

Response:

  • We have addressed this in Lines 108-118 on page 3 in the Data source section as per the reviewer comment in the revised version of the manuscript.
  • Please mention the software used for the statistical analysis.

Response:

  • GeoDa software version 1.2 (this was mentioned in Lines 126-127 on page 3) and R software (this was mentioned in Lines 139-140 on page 3) were used for the statistical analysis.
  • The authors must compare the results of similar studies reported in Ethiopia and Africa region. Please refer the reference 10-18, 21-23, and 37 of the manuscript.

Response:

  • We have considered the comments and addressed them in the discussion section (please see Lines 298-301 and 310-312 on page 8, and Lines 358-364 on page 9).
  1. The authors must discuss the effect of COVID-19 pandemic on the epidemiology of the TB.

Response:

  • We have dedicated a paragraph to address this comment in the discussion section (please see Lines 382-390 on page 10).
  1. A separate paragraph discussing the recommendation based on the result must be mentioned at the end of the discussion.

Response:

  • Recommendations based on the study results are given in the last paragraph of the discussion section (please see Lines 391-396 on page 10).

Reviewer 5 Report

This is very interesting data; the manuscript is well-detailed and structured.

The authors might consider the following comment for improving the manuscript

1-The study design is missed, please add it

2-Result: Please provide the study participant flow chart

3-In general, please avoid repeating the result clearly shown in the figures 

Author Response

Reviewer 5

Comments and Suggestions for Authors

This is very interesting data; the manuscript is well-detailed and structured.

The authors might consider the following comment for improving the manuscript

 1-The study design is missed, please add it

Response:

Thank you for the comments

  • We have used a secondary data for this study. This and methodology used for the analysis discussed in Material and Methodology section.

2-Result: Please provide the study participant flow chart

Response:

  • The study is based on yearly aggregated TB and HIV case notifications data collected at district levels. Therefore, this makes it difficult to prepare study participants flow chart as participants or patients’ details are not available in the data that were provided to the authors.

3-In general, please avoid repeating the result clearly shown in the figures 

Response:

  • The authors tried their best to avoid such reputation in the revised version.

Round 2

Reviewer 4 Report

The manuscript can be accepted 

Author Response

We thank the reviewer for the above comment. We have considered the comment and revised our manuscript.

Please see the changes made in the revised manuscript, highlighted in blue color.